# Crystal Structure of the Human Copper Chaperone ATOX1 Bound to Zinc Ion

**DOI:** 10.3390/biom12101494

**Published:** 2022-10-16

**Authors:** Vincenzo Mangini, Benny Danilo Belviso, Maria Incoronata Nardella, Giovanni Natile, Fabio Arnesano, Rocco Caliandro

**Affiliations:** 1Institute of Crystallography, CNR, Via G. Amendola 122/o, 70126 Bari, Italy; 2Department of Chemistry, University of Bari “Aldo Moro”, Via E. Orabona 4, 70125 Bari, Italy

**Keywords:** X-ray crystallography, molecular structure, metal ions, zinc, copper transport proteins, metallochaperones, Atox1

## Abstract

The bioavailability of copper (Cu) in human cells may depend on a complex interplay with zinc (Zn) ions. We investigated the ability of the Zn ion to target the human Cu-chaperone Atox1, a small cytosolic protein capable of anchoring Cu(I), by a conserved surface-exposed Cys-X-X-Cys (CXXC) motif, and deliver it to Cu-transporting ATPases in the trans-Golgi network. The crystal structure of Atox1 loaded with Zn displays the metal ion bridging the CXXC motifs of two Atox1 molecules in a homodimer. The identity and location of the Zn ion were confirmed through the anomalous scattering of the metal by collecting X-ray diffraction data near the Zn K-edge. Furthermore, soaking experiments of the Zn-loaded Atox1 crystals with a strong chelating agent, such as EDTA, caused only limited removal of the metal ion from the tetrahedral coordination cage, suggesting a potential role of Atox1 in Zn metabolism and, more generally, that Cu and Zn transport mechanisms could be interlocked in human cells.

## 1. Introduction

Zinc (Zn) is an essential element in living organisms and the mechanism of how cells ensure its proper allocation to specific metalloproteins is a topic of current interest. Zn ion serves as catalytic or structural cofactor for a plethora of different proteins, which are located in the cytoplasm and in many organelles of eukaryotic cells, including the nucleus, endoplasmic reticulum, Golgi apparatus, secretory vesicles and mitochondria [1,2]. Evolution has selected effective mechanisms for transport, storage, and distribution of Zn ion within the cell. Cellular import/export of Zn ions is tightly controlled by the ZIP and ZnT families of proteins that regulate the Zn fluxes across membranes [3]. The buffering of free Zn ions is entrusted to metallothioneins, whilst Zn excess is stored in subcellular organelles, such as vesicles [4]. Alteration in the Zn homeostasis can cause serious health problems, such as in the case of Zn deficiency, which, according to the World Health Organization, can affect from 4 to 73% of the population depending upon the geographical areas [5]. Zn deficiency, due to inadequate dietary intakes or physiological unbalances, can also lead to harmful diseases, such as cancer [6,7,8], diabetes [9,10,11], neurodegeneration [12,13], autoimmunity [14], etc. [4,15].

The strong relationship between Zn dyshomeostasis and serious chronic diseases asks for increasing efforts in understanding the mechanisms by which the level of Zn is tightly regulated in the cell and how Zn is transferred to specific metalloproteins.

An intriguing possibility is that the transport mechanisms of Zn and copper (Cu) can be closely interconnected, although the two essential metal ions appear to differ in their functions. Cu is primarily utilized for its redox activity, which also contributes to the toxicity of this metal [16], whereas the Zn ion is considered redox-inactive, but can affect several cellular redox processes [17]. With respect to Cu, Zn is more abundant in biological systems (the Zn:Cu molar ratio ranging between 10^8^ and 10^12^ [18,19]) and exerts a structural/catalytic role in a much greater number of proteins [20]. Nevertheless, the intracellular availability of both metals has to be carefully controlled [21]. Another important difference between these two metal ions is the delivery mechanism. Cu trafficking generally involves specific metallochaperones that transport and deliver the metal ion to its targets [21,22]. In contrast, in the case of Zn, the most accepted hypothesis is that metallothioneins involved in Zn storage can also act as a direct source of Zn for the target proteins [23,24]. However, recent evidence indicates that ZNG1 can use GTP hydrolysis to deliver Zn^2+^ ions to the catalytic sites of target enzymes [25].

The discovery that Cu and Zn metabolisms are tightly connected has led to the approval by the US FDA of Zn acetate administration as maintenance therapy for patients affected by Wilson disease, a disorder of Cu metabolism leading to Cu overload [26]. Such an effect may be related to the Zn-induced overexpression of intestinal metallothioneins that bind Cu very tightly [27]. Moreover, it is also possible that administration of Zn can prevent the reabsorption of endogenously secreted Cu, leading to a substantial negative Cu balance in patients with Wilson disease. In *Chlamydomonas,* it has also been demonstrated that an excess of Zn can induce Cu deficiency [28], whereas, under Zn starvation, Cu is trapped in lysosome-related organelles (called cuprosomes). Extended X-ray absorption fine structure (EXAFS) measurements indicate that, in storage vesicles, Zn [29] and Cu [30] share a similar coordination environment, with two N and one S-donor ligands. Furthermore, Cu transporter 1 (Ctr1) transcripts are abundant in Zn-limited cells, supporting the notion that Cu hyperaccumulation is driven by high Ctr1 expression. When the Zn levels return to normal range, Cu is released with simultaneous trapping of Zn [30]; however, the driving force for moving Cu in and out of these traps is still unknown. All these data indicate that Cu bioavailability might depend on a complex interplay with Zn that involves metal binding motifs of Cu transporters.

Based on the above-mentioned considerations, we have investigated the Zn ion targeting of proteins involved in Cu homeostasis, focusing our attention on the structural aspects of the interaction between Zn and the human Cu-chaperone antioxidant-1 (Atox1). Atox1 is a cytosolic small protein (68 amino acids) able to chelate Cu by a conserved surface-exposed Cys-X-X-Cys (CXXC) motif [22,31,32]. Once bound to Atox1, Cu is delivered to cytoplasmic metal-binding domains of ATP7A and ATP7B (also called Menkes and Wilson disease proteins, respectively), two homologous multi-domain P_1B_-type ATPases located in the trans-Golgi network, for incorporation into Cu-dependent enzymes. The structure of human Atox1 has been extensively investigated and the structures of the protein loaded with various metal ions, such as copper [33], cadmium [33], mercury [33], and platinum [34] (PDB ID: 1FEE, 1FE0, 1FE4, and 4QOT, respectively), have been determined by X-ray diffraction. However, high-resolution atomic models of Atox1 loaded with Zn ions have not been reported so far. Badarau and colleagues investigated the crosstalk between Zn and Cu in cyanobacteria and obtained the structure of the Cu-chaperone Atx1 from cyanobacterium *Synechocystis* loaded with Zn ions, gaining insight into the Zn and Cu binding motifs [35,36].

In this study, we report the first structure of the human Cu chaperone Atox1 loaded with Zn ion, highlighting how the protein could also be involved in human Zn metabolism and, more generally, how the transport mechanisms of Cu and Zn could be closely interconnected in human cells.

## 2. Materials and Methods

### 2.1. Atox1 Expression and Purification

BL21(DE3)Gold *Escherichia coli* cells containing pET21a with the gene encoding Atox1 were grown at 37 °C to an optical density of 0.5 at 600 nm; then, the protein expression was induced with 1 mM isopropyl β-D-1-thiogalactopyranoside. After 4 h of induction, the cells were harvested and centrifuged for 20 min at 4 °C. The cellular pellet was suspended in 20 mM 2-(N-morpholino)ethanesulfonic acid (MES; pH 5.5), 1 mM ethylenediaminetetraacetic acid (EDTA), 5 mM dithiothreitol (DTT), and 1 mM phenylmethylsulfonyl chloride (protease inhibitor), and it was lysed by sonication. The cellular lysate was centrifuged at 20,000 rpm at 4 °C for 1 h. The supernatant, containing the recombinant protein, was loaded onto a HiPrep Sepharose Fast Flow cation exchange column (SP FF 16/10, GE Healthcare, Chicago, IL, USA) pre-equilibrated with 20 mM MES, 5 mM DTT and 5 mM EDTA (pH 5.5), and eluted with a column volume gradient of 20 mM MES (pH 5.5), 5 mM DTT, 5 mM EDTA and 500 mM NaCl. Atox1-containing fractions, eluted at ∼200 mM NaCl, were concentrated and further purified on a size-exclusion column (Superdex 75 10/300 GL, GE Healthcare, Chicago, IL, USA) equilibrated with 20 mM MES, 5 mM DTT, 5 mM EDTA, and 150 mM NaCl (pH 6.0). Fractions containing pure Atox1 were pooled and washed with 25 mM sodium phosphate and 2 mM DTT (pH 7.0), by using Amicon Ultra centrifugal filters with 3 kDa cutoff (Millipore, Burlington, VT, USA). The protein sample was concentrated to 1.5 mg mL^−1^ (protein concentration determined by UV-visible absorbance at 280 nm) and stored at −20 °C. All purification steps of Atox1 were performed in the presence of 5 mM DTT as reducing agent to preserve the protein in its active form by keeping reduced its cysteine residues. The purity of the protein was determined by sodium dodecyl sulfate-polyacrylamide gel electrophoresis (SDS-PAGE) and electrospray ionization mass spectrometry (ESI-MS).

### 2.2. Crystallization and Structure Determination of the Zn-Atox1 Adduct

Atox1 samples were concentrated to 12.5 mg mL^−1^ and treated with different amounts of ZnSO_4_, previously dissolved in 25 mM sodium phosphate and 2 mM DTT (pH 7.0), in order to obtain 1:0.5 and 1:1 protein/metal ion ratio. The reaction solution was incubated at 20 °C for 1 h and used for the crystallization. Crystallization experiments were performed at 20 °C by using the sitting drop technique: 2 µL of protein solution (concentration ranging from 12.5 to 10 mg/mL) was mixed with 2 µL of the reservoir solution made of Li_2_SO_4_ (concentration ranging from 1.8 to 2.05 M), MES 100 mM at pH 6.5, and 2.5% glycerol. Needle-shaped crystals were obtained by equilibration of the drop against the reservoir solution for one week. Some Zn-Atox1 crystals were put in a soaking solution containing ethylenediaminetetraacetic acid (EDTA) at 2 mM concentration. Data collections on crystals were made at the European Synchrotron Radiation Facility (ESRF). The X-ray diffraction experiment was performed at 100 K at the microfocused beamline ID-23-1 equipped with a PILATUS 6M-F detector (Dectris AG, Baden-Daettwil, Switzerland). To enable the detection of Zn or Cu ions bound to the protein through anomalous diffusion, data were taken at X-ray beam energies of 9090 and 9690 eV (the energies related to the K transition absorption for Cu and Zn, respectively), by using different crystals of the same drop and/or pointing the beam at different points of the same crystal. Data processing has been performed by autoPROC toolbox [37], which exploits the XDS program [38] for data reduction, the POINTLESS program [39] for space group assignment, and the AIMLESS program [40] for data scaling and merging. The structure was solved by molecular replacement using the REMO program [41,42] included in SIR2014 (v. 17.01) [43]. The crystal structure with PDB code 4QOT [34], having 100% sequence identity, deprived of heavy metal ions and water molecules, was used as a molecular replacement model. The matching between experimental and calculated electron densities was improved by performing an automatic building procedure on the structure obtained by molecular replacement in “rebuilt-in-place” mode by using Autobuild wizard [44] included in the PHENIX crystallographic structural solution suite. Experimental phase information has been exploited to confirm the position of the heavy metals by using the SAD target function in the REFMAC software (v. 5.032) of the CCP4 crystallographic package (v. 7.0.078) [45]. The Zn-S bond distance was restrained to 2.34 Å (ideal distance suggested by REFMAC software) to avoid the collapse in the high electron density of the metal ion of the cysteine residues of the CXXC motif. The refinement procedure of the whole structure was performed using the REFINE software [46] included in the PHENIX crystallographic structural solution suite (v. 1.20.14487). The COOT program (v. 0.8.9.2) [44] was used to complete the crystal structure model by taking into account the electron density maps 2Fo-Fc and Fo-Fc. Finally, the crystal structure was validated by using the MolProbity validation tool [47] and deposited in the Protein Data Bank under PDB code 7ZC3.

## 3. Results and Discussion

Atox1, co-crystallized with different amounts of ZnSO_4_ (1:0.5 and 1:1 protein/metal ion ratio) yielded needle-shaped crystals, which were bigger in size for the 1:1 protein/metal ratio (Figure 1). Data collections were made on needle-shaped crystals of about 50 µm.

The structure (PDB ID code 7ZC3) was solved to a resolution of 1.9 Å, which is close to that reported for Atox1 dimers crystallized in the presence of copper or platinum ions, ranging from 1.75 to 1.85 Å [33,34]. The crystals exhibit P6_5_ symmetry and two protein molecules per asymmetric unit (hereinafter named chains A and B). The two chains are equally folded and arranged in a two-fold axis passing through the cage of the metal binding motif (Figure 2), similar to what was found in the previously solved dimeric structures [34]. The main crystallographic parameters are reported in Table 1.

A positive Fo-Fc electron density signal was found between the two protein chains near the Cys12 and Cys15 residues, supporting the presence of a metal ion (Figure 3a). However, the identity of this metal ion could remain uncertain, although exogenous Zn was added to the crystallization solution. In fact, it is well known that Atox1 has a high affinity for the physiological metal Cu, for which a very low dissociation constant has been determined (K_D_ < 10^−17^ M) [48]. Therefore, adventitious Cu could efficiently compete with exogenous Zn (added in the crystallization procedure) for coordination to the protein. Cu and Zn affinity constants were determined by Dennison’s group for Atx1, the Atox1 homologue in cyanobacteria, by competition experiments. The Cu affinity for the monomeric protein (K_b_ = 4.7 × 10^17^ M^−1^) was found to be several orders of magnitude greater than that of Zn (K_b_ = 7.2 × 10^8^ M^−1^) [49]. As a consequence of such a large affinity for Cu, a recently reported structure of Atox1 contained Cu (PDB ID: 4YEA) [50] instead of platinum (PDB ID: 3IWX), although cisplatin had been added to the crystallization sample [51]. Fortunately, it is possible to confirm the identity and location of a metal ion within a protein by collecting X-ray diffraction data at photon energies close to the metal absorption edge. In our case, a peak in the anomalous scattering density map close to the cysteine residues of the CXXC motifs was found at 9690 eV (Figure 3b), an energy right above the Zn K-edge (9640 eV), while no anomalous signal was found close to the Cu K-edge (8970 eV), as shown in Figure 3c. X-ray diffraction data collected at 8970 eV are included in the Supporting Data. Therefore, it is possible to conclude that the electron density found in between the CXXC metal binding motifs of the two chains belongs to a Zn ion. Zn binds to the sulfur atoms of Cys12 and Cys15 of the two chains in a tetrahedral arrangement (Figure 3b), similarly to the Atox1 homodimers formed with other metal ions. Moreover, the present structure is in full agreement with that of Atx1 reported by Badarau and colleagues [35]. It should be emphasized that, unlike Atox1, Atx1 contains an unusual His in position 61 (Lys in the case of human Atox1), which is also a metal-binding ligand and is supposed to further stabilize the dimeric form of the protein in solution. Additionally, we have confirmed the identity and location of the Zn metal ion in the Atox1 structure via anomalous scattering experiments above the Zn K-edge and between the Cu K-edge and the Zn K-edge, thus overcoming a possible assignment ambiguity with the Cu ion.

Comparison of the present structure with those of other Atox1 dimers containing different metal ions (PDB ID: 1FEE [33], 1FE0 [33], 1FE4 [33], and 4QOT [34]) indicates that the protein backbones are similar, with RMSDs of Cα atoms (calculated by superposition of the homodimer present in the asymmetric unit) as low as 0.178, 0.233, 0.350, and 0.157 Å, respectively. The superposition of the PDB structures containing an Atox1 dimer is reported in Figure 4 and confirms that there are no significant backbone displacements.

The Zn ion presents a crystallographic occupancy of 0.69 and is surrounded by four cysteine residues (Cys12 and Cys15 of chains A and B). A partial crystallographic occupancy of the metal-binding site is reported also in other deposited Atox1 dimeric structures (PDB ID: 1FEE, 1FE0, 1FE4, and 4QOT), with values of 0.90 (Cu), 1.00 (Cd), 0.71 (Hg), and 0.35 (Cu)–0.65 (Pt), respectively. The sulfur atoms of the cysteines around the cage are at 2.3–2.4 Å from the metal center and arranged in a tetrahedral geometry, with bond angles ranging from 105° to 118°, as expected for a regular tetrahedral coordination environment.

The Zn-Atox1 crystals were also soaked for 10 min with 2 mM EDTA, a powerful Zn-chelator, which has an affinity constant several orders of magnitude greater than that of Atox1 towards Zn^2+^ [52] and is normally used to test the affinity of proteins for metal ions in biological studies. After soaking, the crystal was subjected to an X-ray diffraction experiment and the structure solved to a resolution of 1.7 Å. It was found that the Zn ion keeps the same coordination but the crystallographic occupancy decreases to 0.59 (Figure 5).

## 4. Conclusions

The aim of this investigation was to prove the interaction between the Zn ion and the human Cu-chaperone Atox1. For the first time, crystals of Zn-bound human Cu-chaperone Atox1 have been obtained and the structure solved by X-ray diffraction to a resolution of 1.9 Å. Moreover, although the affinity of Atox1 for Zn^2+^ is lower than that of EDTA, soaking of the Zn-loaded Atox1 crystals with 2 mM EDTA for 10 min resulted in only limited removal of the metal ion from the coordination cage. Although other phenomena, such as crystal packing, diffusion of EDTA in the crystal and the kinetics of Zn exchange between the cysteine cage and EDTA, may contribute to the stability of the Zn-Atox1 dimer in the solid state, it appears that the CXXC motif of Atox1 can tightly bind a Zn^2+^ ion in a homodimer and, owing to the high level of Zn in the cell, the Zn-Atox1 binding could play a role in modulating the binding of Cu to its target proteins.

To summarize, this investigation indicates that Atox1 may be involved in Zn metabolism and that, more generally, Cu and Zn transport mechanisms may be closely related. These findings can also contribute to a better understanding of diseases caused by defective metal homeostasis in mammals.

Many questions remain to be answered in order to obtain a clear picture of the relationship between Zn and Cu transport proteins. Furthermore, this type of investigation needs to be extended to other Cu-binding proteins, including multi-domain proteins, such as Cu-ATPases.

## Figures and Tables

**Figure 1 biomolecules-12-01494-f001:**
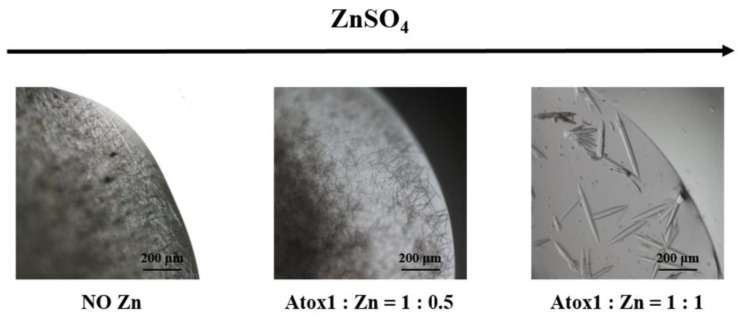
Microscopic images of the crystallization outcomes of Atox1 co-crystallized with increasing amounts of ZnSO_4_.

**Figure 2 biomolecules-12-01494-f002:**
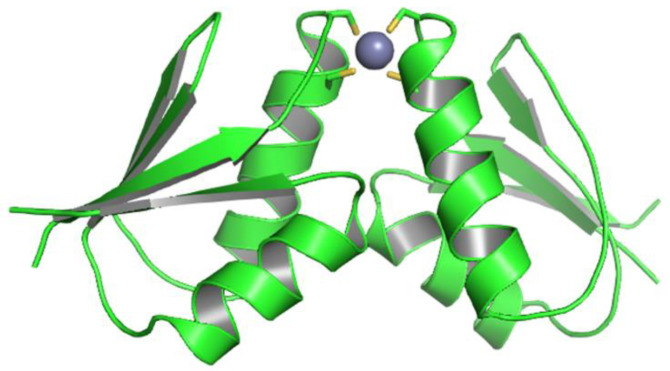
Crystal structure of the Atox1 dimer crystallized in the presence of ZnSO_4_ (PDB code 7ZC3). The Atox1 molecules are shown in cartoon representation and the metal ion as a gray sphere. Protein residues within 3 Å from the metal are shown as sticks, with C and S atoms colored in green and yellow, respectively.

**Figure 3 biomolecules-12-01494-f003:**
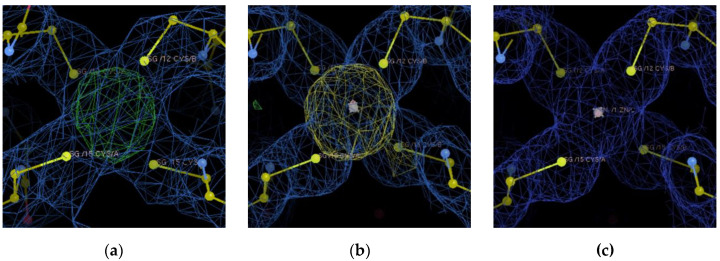
Metal binding site within the cage formed by the CXXC motifs of the two Atox1 protein chains in the asymmetric unit. (**a**) The 2Fo-Fc electron density map (blue, 1.5σ) and the Fo-Fc difference map (green +3σ and red −3σ) around the cage are shown. (**b**) Metal binding site with the anomalous map (yellow 3.5σ) obtained at 9690 eV, 50 eV above the Zn K-edge. (**c**) Metal binding site with the anomalous map (yellow 3.5σ) obtained at 9090 eV, between the Cu K-edge and the Zn K-edge.

**Figure 4 biomolecules-12-01494-f004:**
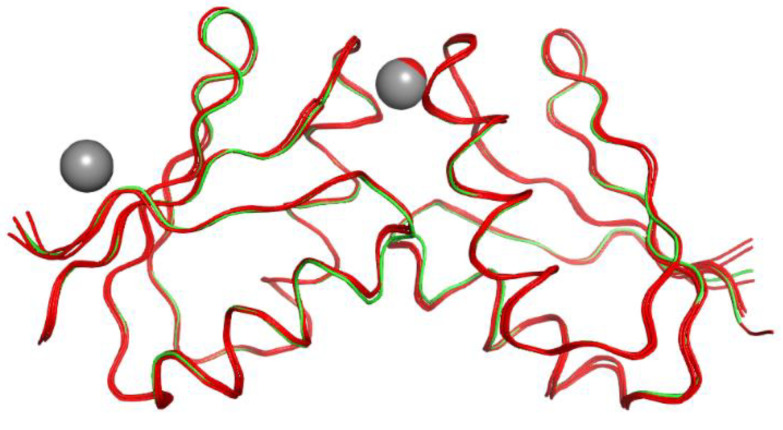
Superposition of the crystal structures containing Atox1 dimers. Protein molecules are in ribbon representation, while metal ions appear as spheres. Previously published 1FEE, 1FE0, 1FE4, and 4QOT are in red color, whilst the present crystal structure 7ZC3 is shown in green. Metal ions are Cu for 1FE0, Cd for 1FEO, Hg for 1FE4, Pt for 4QOT, and Zn for 7ZC3.

**Figure 5 biomolecules-12-01494-f005:**
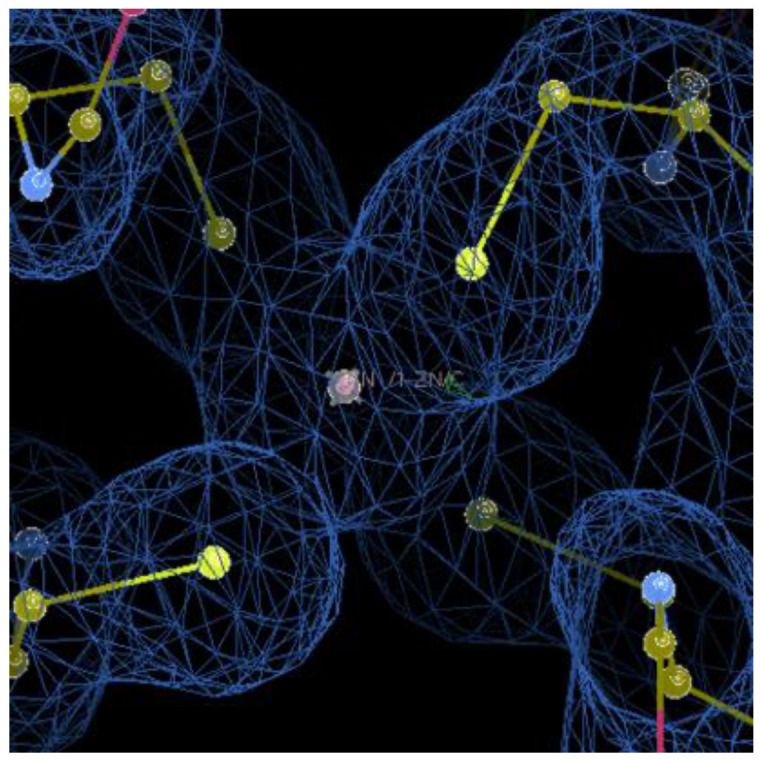
Metal-binding site of Zn-Atox1 crystals soaked with 2 mM EDTA for 10 min. The 2Fo-Fc electron density map (blue, 1.5σ) and Fo-Fc difference map (green +3σ and red −3σ) around the cage are shown.

**Table 1 biomolecules-12-01494-t001:** Diffraction data statistics of crystals obtained from co-crystallization of Atox1 and ZnSO_4_ in 1:1 ratio (PDB code 7ZC3). Values related to the outer high-resolution shell are in brackets.

Parameter	Values
Wavelength (Å)	1.28
Resolution range (Å)	42.53–1.91 (1.95–1.91)
Space group	P6_5_
Unit cell parameters (Å)	
a	78.226
b	78.226
c	54.637
Total number of reflections	81,486 (5587)
Total number of unique reflections	23,876 (968)
<I/σ(I)>	10.4 (2.1)
Half-set correlation CC (1/2)	0.997 (0.711)
R_merge_	0.093 (0.911)
Completeness (%)	99.1 (97.2)
Multiplicity	5.6 (5.6)
Anomalous completeness (%)	92.4 (89.7)
Anomalous multiplicity	2.7 (2.7)
DelAnom CC (1/2)	0.266 (0.044)

## Data Availability

The crystal structure presented in this study is publicly available at the Protein Data Bank under the PDB code 7ZC3.

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
