# Peer review of "Crystal Structure of the Human Copper Chaperone ATOX1 Bound to Zinc Ion"

_biomolecules, 2022, doi:10.3390/biom12101494_

Round 1
Reviewer 1 Report
The manuscript entitled “Crystal structure of the human copper chaperone Atox1 bound to zinc ion” is focused on the structural investigation of the human Cu-chaperone Atox1 in complex with zinc ions, to better understand the interconnection between Cu and Zn in human cells. The authors report the first structural characterization of Atox1 loaded with Zn ions, providing clues on the involvement of this protein in human Zn metabolism and on the potential interconnection between the transport mechanisms of Cu and Zn in human cells. The manuscript includes interesting new results, and I would recommend manuscript publication after addressing the following minor issues.
Minor issues:
1. Introduction, line 87. “to date” could be changed to “so far”.
2. Results and Discussion, line 167. “of other metal ions” is generic, please specify.
3. Table 1. “Hal-set” should be changed to “Half-set”.
4. Results and Discussion, lines 198-199. Please change “an energy which is right above that of the Zn K-edge of other metal ions” to “an energy right above the Zn K-edge of other metal ions”.
5. Results and Discussion, lines 218-221. This sentence should be supported by a figure showing the comparison between the structure of the Zn-Atox1 complex with those of other Atox1 dimers containing copper and/or other metal ions (at least the comparison with the Cu-Atox1 complex should be shown here). The citing references of the structures mentioned in the sentence could also be added here (and in the Introduction, at line 85).
6. Results and Discussion, lines 230-231. The abbreviation EDTA has been introduced at line 102.
Reviewer 2 Report
It is a well written mansucript for an important problem of interoaly of Zn and Cu ion, The author clearly proved the bounding capability of Zn to ATOX1 protein between two well defined motifs of protein. Additionally they guess the stability of bonding using EDTA. The occupancy of site is 0.69 which is smaller than in Cu case (0.9) but the Zn/Cu ratio is about 108-1012. I think it is an important contribution, I have only two coupled question. Have the author try the crystallisation in a mixed Zn..Cu solution. If the Zn/Cu ratio is so high then it can give us a guess for the bonding capability too, due to in the cell the atox1 bond mainly for cu. Can the author comment this?
Reviewer 3 Report
According to this report, X-ray diffraction experiments verify the presence of Zn ions in the crystal structure of the Atox1 homodimer protein. The work is well designed, and the results are convincing. It would be helpful to provide insight into the mechanism behind the substitution of Zn ions for Cu ions. For the author, I have the following questions:
1. Did the author use media that was deficient in Cu ions?
2. Data from anomalous scattering experiments should be included.
3. A unit for the affinity of Atox-1 monomer towards Cu and Zn is missing.
